# Muscle Fiber Composition Changes after Selective Nerve Innervation

**DOI:** 10.3390/ijms23147856

**Published:** 2022-07-16

**Authors:** Shiho Watanabe, Hiroko Ochiai, Hisashi Sakuma, Taisuke Mori, Masaki Yazawa, Aiko Oka, Kazuo Kishi

**Affiliations:** 1Department of Plastic and Reconstructive Surgery, Okayama University Hospital, Okayama 700-8558, Japan; 2Laboratory of Regenerative Medicine, Department of Plastic and Reconstructive Surgery, Division of Hearing and Balance Disorder, National Institute of Sensory Organs, National Hospital Organization Tokyo Medical Center, Tokyo 152-8902, Japan; ochiroko@gmail.com (H.O.); aka10aiko12@gmail.com (A.O.); 3Department of Plastic and Reconstructive Surgery, Tokyo Dental College Ichikawa General Hospital, Chiba 272-8513, Japan; prssakuma@gmail.com; 4Department of Pathology, National Cancer Center Research Institute, Tokyo 104-0045, Japan; tamori@ncc.go.jp; 5Department of Plastic and Reconstructive Surgery, Keio University School of Medicine, Tokyo 160-8582, Japan; yazawa@a7.keio.jp (M.Y.); kkishi@a7.keio.jp (K.K.)

**Keywords:** muscle fiber type, facial paralysis, dualinnervation, masseter muscle

## Abstract

Facial nerve paralysis interferes with mimetic muscle function. To reconstruct natural facial movement, free muscle flaps are transplanted as new muscles. However, it is difficult to maintain resting tonus. A dual innervation technique in which other nerves such as the hypoglossal nerve or contralateral facial nerve are added is often applied. Using 10-week-old rats (*n* = 10), the masseteric and hypoglossal nerves were cut, and the distal stump of the masseteric nerve and the proximal stump of the hypoglossal nerve were then sutured (suture group). In the other group, the masseteric nerve was cut and cauterized (cut group). Immunohistochemistry and microarray were performed on the extracted masseter muscle. The immunohistochemistry results suggested that the muscles in the suture group obtained oxidative characteristics. The microarray showed the genes involved in mitochondrial function, including Perm1. In summary, our data support the validity of the dualinnervation technique for facial paralysis treatment.

## 1. Introduction

Facial nerve paralysis is a condition in which the facial nerve is completely or partially paralyzed, thereby interfering with the function of the mimetic muscles. To reconstruct natural facial movement in patients with facial nerve paralysis, free muscle flaps are transplanted as new mimetic muscles [1,2,3]. The ideal neural motor source for transplanted muscle is the healthy side of the facial nerve, which allows resting tonus and synchronous contraction. However, the small number of regenerative axons often results in weak contraction of the transplanted muscle. To overcome this disadvantage, the masseteric nerve is often used as a neural motor. Although the use of this nerve alone can lead to a rapid and strong recovery of mimetic function, it is difficult to maintain resting tonus. To create more natural facial expressions, we often perform a dual innervation technique [4] in which we add other nerves such as the hypoglossal nerve or contralateral facial nerve, and good results have been obtained. Owing to this phenomenon, we hypothesize that the transplanted muscles undergo a muscle fiber type transition to represent muscle more characteristic of the nerve by which it is reinnervated. Inherently, skeletal muscle contains various fiber types with different contraction velocities and fatigability characteristics. These fiber types can be classified as slow-twitch (type 1) and fast-twitch fibers (type 2) based on their levels of myoglobin, the number of mitochondria, and the expression of glycolytic enzymes. We hypothesize that the dual innervation technique confers the transplanted muscles with both type 1 and type 2 fiber characteristics, allowing them to maintain resting tonus as well as movement, contributing to more natural facial expression. In addition, it is expected from clinical results that this can be accomplished by adding both the hypoglossal/facial and masseteric nerves, which normally serve type 1 and type 2 fibers, respectively.

It has been reported that the muscle fiber type in the lower extremities changes when the innervation is altered. Krysciak et al. reported that neuronal reinnervation in the medial gastrocnemius muscle occurred as ALS progressed and the percentage of fast-twitch muscle increased [5]. Robbins et al. revealed histologically and electrophysiologically that the properties of the guinea pig soleus muscle were altered by cross-nerve innervation [6]. Romanul et al. demonstrated histologically and electrophysiologically that nerve crossing of the flexor digitorum longus and soleus muscles in cats and rats altered their properties [7].

In rat skeletal muscles, type 2 fibers can be further subdivided into the following types: type 2A, fast-twitch oxidative; type 2B, fast-twitch glycolytic; and type 2X, an intermediate type between type 2A and 2B. The superficial layer of the rat masseter muscle is composed mainly of type 2 muscle fibers. In this study, we investigated whether the muscle fiber composition of the rat masseter muscle could be altered by changing the innervation from the masseteric nerve to the hypoglossal nerve by evaluating immunohistochemical findings. We also tried to reveal the nature of the cross-innervated muscle by performing a microarray analysis.

## 2. Results

### 2.1. Histological Findings

Hematoxylin and eosin (H&E) staining revealed the nylon suture that had been used for coaptation. There were no morphologically abnormal findings among the suture group (Figure 1a). We observed neurofilament antibody positivity in the area of the coaptation of the hypoglossal nerve and masseteric nerve (Figure 1b).

The cross-sectional area of the muscle fibers was calculated. The average area of muscle fibers was 4231.20 ± 432.70 µm^2^ (mean ± standard deviation; SD) in the control, 4767.32 ± 695.04 µm^2^ in the suture group, and 3112.00 ± 293.49 µm^2^ in the cut group. The area of muscle fibers in the cut group was significantly smaller than that in the other groups (Figure 2).

In the control group, immunofluorescence staining of the masseter muscle illustrated that almost all specimens were BAF8-negative (Figure 3a), slightly SC71-positive (Figure 3b), and strongly BFF3-positive (Figure 3c). The average percentage of type 1, type 2A, type 2B and the estimated type 2X muscle fibers among the muscle fiber areas was 2.92 ± 2%, 4.40 ± 3.62%, 41.52 ± 6.30%, and 51.16 ± 6.77%, respectively (Figure 3d). This indicated that the muscles were composed mainly of type 2 fibers, especially type 2B fast-twitch glycolytic muscle fibers.

Meanwhile, the specimens of the suture group were slightly BAF8-positive (Figure 4a) and slightly SC71-positive (Figure 4b). All specimens of the suture group were BAF8-positive, and the average percentage of type 1, type 2A, type 2B and the estimated type 2X muscle fibers was 13.50 ± 2.83%, 9.38 ± 2.01%, 52.17 ± 8.30%, and 25.00 ± 7.72%, respectively.

In the cut group, most specimens featured an area that was negative for BAF8, SC71 and BFF3 (Figure 5a–c). The average percentages of type 1, type 2A, type 2B and the estimated type 2X muscle fibers were 0.92 ± 0.87%, 6.67 ± 6.65%, 27.99% ± 10.80%, and 64.42 ± 7.68%, respectively (Figure 5d).

By statistically comparing the percentage of each muscle fiber type in each group, it was revealed that the suture group showed a significantly higher percentage of BAF8 positivity than the other two groups (Figure 6a). As for BFF3, the cut group revealed significantly lower positivity than the suture group (Figure 6b). The negative area for BAF8, SC71, and BFF3 was significantly smaller in the suture group than in the other two groups (Figure 6c).

### 2.2. Microarray

#### 2.2.1. Gene Expression in Relation to the Muscle Fiber Type Composition

Compared to the findings in the control group, 310 genes were upregulated and 928 genes were downregulated in the suture group. Compared to the findings in the cut group, 628 genes were upregulated and 1021 genes were downregulated in the suture group.

Regarding the 404 proteins related to the muscle fiber type as detected by Murgia et al. [8], 30, 2, 41, 104, and 151 proteins were oxidative muscle-specific, intermediate type-specific, glycolytic muscle-specific, oxidative muscle-dominant, and glycolytic muscle-dominant, respectively, and the remaining 76 proteins were non-specific. We examined the list of proteins for which genes were up- or downregulated in the suture group. In total, 28 of 328 genes (8.5%) were upregulated oxidative muscle-related genes or downregulated glycolytic muscle-related genes, whereas 38 genes (11.6%) were downregulated oxidative muscle-related genes or upregulated glycolytic muscle-related genes. We focused on the genes that were differentially expressed in the suture group relative to both the cut and control groups, and nine genes were upregulated oxidative muscle-related or downregulated glycolytic muscle-related genes (upregulated oxidative muscle-related genes, Lrrc39 (leucine-rich protein), myomasp (myosin-interacting, M-band-associated stress-responsive protein), Ckmt2 (creatine kinase, mitochondrial 2), Ndufs7 (NADH: ubiquinone oxidoreductase core subunit S7), and Perm1 (PPARGC1 and ESRR-induced regulator, muscle 1); downregulated glycolytic muscle-related genes, Ddah1 (dimethylarginine dimethylaminohydrolase 1), Anxa5 (annexin A5), Hsp90aa1 (heat shock protein 90 alpha family class A member 1), Ahnak2 (AHNAK nucleoprotein 2), and P4hb (prolyl 4-hydroxylase subunit beta)). Meanwhile, another nine genes were downregulated oxidative muscle-related genes or upregulated glycolytic muscle-related genes (upregulated glycolytic muscle-related genes, Gpd2 (glycerol-3-phosphate dehydrogenase 2), Phkb (phosphorylase b kinase regulatory subunit), Phkg1 (phosphorylase kinase catalytic subunit gamma 1), and Jph2 (junctophilin 2); downregulated oxidative muscle-related genes, Casq2 (calsequestrin-2), Myoz2 (myozenin 2), Tnni1 (troponin I, slow skeletal muscle), Sptlc2 (serine palmitoyltransferase long chain base subunit 2), and Ankrd2 (ankyrin repeat domain 2); Table 1).

#### 2.2.2. PPI Network Analysis

Suture group vs. control group: the PPI network constructed for the genes upregulated in the suture group versus the control group (262 nodes and 263 edges) revealed that the most functional module consisted of Ndufs7, Ndufb4 (NADH: ubiquinone oxidoreductase subunit B4), Mpc1 (mitochondrial pyruvate carrier 1), Mpc2 (mitochondrial pyruvate carrier 2), Chchd3 (coiled-coli-helix-coiled-coli-helix domain containing protein 3), and Chchd10 (coiled-coli-helix-coiled-coli-helix domain containing protein 10). The top five hub genes were Suclg1 (succinate–CoA ligase (ADP/GDP-forming) subunit alpha), Gpd2 (glycerol-3-phosphate dehydrogenase 2, mitochondrial), Sucla2 (succinate–CoA ligase (ADP-forming) subunit beta), Aco2 (aconitate hydratase, mitochondrial), and Echs1 (enoyl-CoA hydratase, mitochondrial).

Suture group vs. cut group: the PPI network constructed for the genes upregulated in the suture group compared to the cut group (529 nodes and 1170 edges) illustrated that the most functional module consisted of Perm1, Fbxo40 (F-box protein 40), Mettl21cl1 (methyltransferase-like 21C-like 1), Mylk4 (myosin light chain kinase family, member 4), Smpx (small muscle protein, x-linked), Klhl31 (Kelch-like family member 31), and Lsmem1 (leucine-rich single-pass membrane protein 1). The top five hub genes were Actb (actin, cytoplasmic 1), Stat1 (signal transducer and activator of transcription 1), Esr1 (estrogen receptor 1), Ccnd1 (G1/S-specific cyclin-D1), and Ccl2 (C-C motif chemokine 2).

## 3. Discussion

In this study, we examined whether the characteristics of the newly innervated masseter muscle were changed by suturing the hypoglossal nerve to the masseter nerve. Immunostaining confirmed that the nerve sutures resulted in living continuous nerve fibers. First, the cross-sectional area of the muscle fibers in each group was analyzed, and it was found that the cross-sectional area of muscle fibers was significantly smaller in the cut group than that in the other two groups; this was considered a result of muscle atrophy. On the other hand, the size of the muscle fibers in the control and suture groups was not significantly different; this could be because the muscles in the suture group were successfully innervated by the hypoglossal nerve. Regarding muscle properties, the control group seemed to be composed mainly of type 2B and type 2X with almost no type 1 muscle fibers, supporting the originally known characteristics of the masseter muscles, which are fast-twitch muscles [9,10]. Conversely, statistical analysis revealed that the BAF8-positive area in the masseter muscle in the suture group was significantly larger than that in the other two groups; this could be a strong indication that some muscle fibers switched to a slow-twitch type due to the innervation of the hypoglossal nerve. As the estimated area for type 2X was significantly smaller in the suture group, there is a possibility that the type 2X fibers were converted to type 1. There was no significant difference in the percentage of type 1, type 2A, type 2B, and the estimated type 2X fibers between the control and the cut group. However, this did not necessarily indicate that the muscle fiber composition of the cut group was similar to that of the control group. It is known that a denervated muscle often consists of hybrid muscle fibers [11], suggesting there was an increase in them due to the operation.

Microarray analysis demonstrated that the same number of genes characteristic for oxidative/glycolytic muscle were changed following cross-innervation. However, when analyzing the results in detail, mitochondrial markers such as Ckmt2 and Ndufs7 were upregulated whereas PPI network analysis illustrated that energy production was apparently activated in mitochondria, suggesting that cross-innervation may have caused the muscle fibers to transition to a more oxidative-like state. Skeletal muscle fiber types are classified by the number of mitochondria as well as differences in mitochondrial metabolism and structural characteristics [12,13,14].

Type 1 and type 2A fibers are mitochondria-rich fibers that derive their energy mainly from the oxidative phosphorylation system (OXPHOS), which is present at the mitochondrial inner membrane and is composed of five enzymes [15], namely, NADH: ubiquinone reductase, succinate dehydrogenase, quinol-cytochrome c reductase, cytochrome c oxidase, and ATP synthase. Contrarily, mitochondria-poor type 2B and type 2X fibers produce ATP mainly through anaerobic glycolysis. The top upregulated hub genes in the suture group compared to the control group, specifically Suclg1, Sucla2, Aco2, and Echs1, are also OXPHOS genes, and thus, the results support the possibility that the masseter muscle transitioned from type 2B to type 1 or type 2A, which would support the hypothesis.

In addition, Perm1, which was upregulated in the suture group compared to its expression in both the control and cut groups, is known to induce changes in muscle fiber type without causing histological changes [15]. Cho et al. suggested that Perm1-upregulated muscles displayed a comparable increase in OXPHOS complex I–IV enzyme activity. Moreover, Perm1 had significantly elevated the levels of other mitochondrial enzymes important for oxidative metabolism. This suggested that the mechanism by which Perm1 drives mitochondrial biogenesis involves the regulation of Ca^2+^/calmodulin-dependent protein kinase II (CaMKII), resulting in enhanced myocyte enhancer factor-2 (MEF2) transcription. Indeed, in this study, CaMKII was upregulated in the suture group compared with the control and cut groups. CaMKII-activated MEF2 can induce multiple metabolic targets, including PGC-1a (peroxisome proliferator-activated receptor gamma coactivator 1-alpha), which drives mitochondrial biogenesis.

Nehrer-Taylor et al. revealed that cross-nerve innervation can be used to transform muscle morphology and that slow-twitch fibers contribute to tonus maintenance at rest [16]. Therefore, in combination with the results of the present study, it may be possible to reproduce tonus by switching masseter muscle nerve innervation, thereby converting it to a muscle type similar to that of the facial muscles. As previously mentioned, various muscles are used as materials for muscle transplantation, but in selecting the neural motor source for muscle transplantation, the nerve type selected for innervation appears more important than the muscle itself.

This study had a few limitations. First, the number of examined samples was small, and larger sample size should be considered in the future. Second, an artificial nerve was used for nerve sutures, and the difference between nerve-to-nerve sutures has not been studied. Third, the area of type 2X muscle fibers was estimated by the negative area for BAF8, SC71, and BFF3 antibodies, which may include the area of the hybrid types of muscle fibers; therefore, an additional study is necessary for the future. Fourth, D’Amico et al. suggested that muscle fiber composition differed between sexes. They reported that the soleus and extensor digitorum longus muscles of female mice showed more slow-twitch type dominant patterns than those of male mice. [17] Therefore, the results of facial nerve paralysis surgery may differ between the sexes, and future studies with female rats should be considered. Finally, samples were harvested 2 months after surgery, and a longer follow-up period should be considered.

## 4. Materials and Methods

Fourteen 10-week-old Sprague–Dawley rats (CLEA Japan Inc., Tokyo, Japan) were used in the study. Three rats died during the operation, and one rat was euthanatized because of insufficient feeding attributable to malocclusion during the surgery. Finally, 10 rats were included in the analysis.

### 4.1. Surgery

Before surgery, rats were anesthetized with ketamine hydrochloride (10 mg per 100 g body weight, intraperitoneally). The skin incision was made from the preauricular area to the middle of the neck. The masseter muscle was dissected along the mandible to expose the masseteric nerve. After that, the digastric muscle was resected between the anterior and posterior belly and then flipped, and the sublingual nerve was dissected laterally to the geniohyoid muscle. We categorized animals into the control, suture, or cut groups according to their neural processing. In the suture group, we cut the masseteric and hypoglossal nerves and then sutured the distal stump of the masseteric nerve and the proximal stump of the hypoglossal nerve (*n* = 6; Figure 7 and Figure 8). We used an artificial nerve as a bridge (Renerve, Nipro, Osaka, Japan). In the cut group, we cut and cauterized the masseteric nerve (*n* = 4; Figure 8). We performed the operation unilaterally so that the rats could eat. We confirmed all rats had gained weight at 2 months postoperatively to make sure of their food intake. In the control group (*n* = 6), no surgery was performed. (Figure 8) Rats were euthanized via an overdose of ketamine hydrochloride 2 months after the operation. We resected a 10 × 10 × 10 mm^3^ sample of the bilateral superficial masseter muscle from each rat. We used a 1 × 1 mm^2^ section as the specimen for microarray analysis, and the remaining tissue was used for immunohistochemistry. The superficial masseter muscles of the intact side were used as a control.

### 4.2. Histological Analysis

All immunohistochemical experiments were performed using fresh frozen muscles. The collected muscles were frozen using isopentane chilled over dry ice and stored at −80 °C. Transverse 7 μm serial sections were cut at −20 °C in a cryostat (Leica, Wetzlar, Germany) and placed onto glass slides. For morphological evaluation, H&E staining was performed for each specimen. For immunofluorescence staining, the sections were blocked with phosphate-buffered saline +0.2% Triton X-100 (PBST) and 5% bovine serum albumin (BSA) for 45 min. After that, primary antibodies were diluted in 0.2% PBST and 5% BSA and applied to the slides. BAF8 antibody (Development Studies Hybridoma Bank, Iowa City, IA, USA) recognizes myosin heavy chain type 1, SC71 antibody (Development Studies Hybridoma Bank) recognizes myosin heavy chain type 2A, and BFF3 antibody (Development Studies Hybridoma Bank) recognizes myosin heavy chain type 2B. A secondary antibody (Alexa Fluor-594, Thermo Fisher Scientific, Waltham, MA, USA) was added to each section for 60 min. In addition, a rapid 4′,6-diamidino-2-phenylindole nuclear staining kit (NucBlue, Life Technologies, Thermo Fisher Scientific) was used to improve automatic focusing. All samples were stained in batches according to good laboratory practice, each contained three sections of the same specimen.

We also histologically observed the masseteric and hypoglossal nerves, which were sutured in the suture group, using H&E staining for morphological evaluation. Immunofluorescence analysis was performed using anti-neurofilament M (145kDa) antibody, C-terminus (Merck Millipore, Burlington, MA, USA) as the primary antibody.

An FV3000 Scanning Confocal Microscope (Olympus, Tokyo, Japan) was used for observation. Images were obtained through Fluoview Fv31s Sw software (Olympus, Tokyo, Japan). For the analysis of the cross-sectional area of muscle fibers, Image J software (version 1.53, National Institutes of Health, Bethesda, MD, USA) was used [18,19]. Four1000 × 1000 μm^2^ images were obtained from each muscle slide, and the average area was calculated based on a previous study [20]. Images were opened by the software and converted into 8-bit, and a threshold was used based on the presets for each channel. Next, Gaussian blur was applied to smooth the edges and facilitate fiber identification. Then, the images were converted into binary using the “make binary” command, and “watershed” was applied to separate touching fibers. Individual fibers were counted using the command “analyze particles,” and the results listing the properties of each specimen in consecutive order were shown. To analyze the percentage of muscle fibers positive for each antibody, four 1000 × 1000 μm images were acquired from each group, and the same image processing of cross-sectional areas was performed. As an approximation of type 2X expression, the sum of the negative area percentages for BAF8, SC71, and BFF3 antibodies was calculated in each specimen. The calculated areas and percentages were analyzed with a one-way analysis of variance with a post hoc test (Tukey Kramer test), and the results were considered significant when *p* values < 0.05. Statistical analysis was performed using EZR (Saitama Medical Center, Jichi Medical University, Saitama, Japan), which is a graphical user interface for R (The R Foundation for Statistical Computing, Vienna, Austria). More precisely, it is a modified version of R commander designed to add statistical functions frequently used in biostatistics [21].

### 4.3. Microarray

Total RNA was isolated from the masseter muscle using TRizol™ Reagent (Thermo Fisher Scientific) and quantified using a NanoDrop-2000c spectrophotometer (Thermo Fisher Scientific), and quality was monitored using an Agilent 2100 Bioanalyzer (Agilent Technologies, Santa Clara, CA, USA). Biotinylated ss-cDNA was prepared according to the standard Affymetrix protocol from 100 ng of total RNA (GeneChip WT PLUS Reagent Kit, No. 902281, Thermo Fisher Scientific). Biotinylated ss-cDNA yields were checked using a NanoDrop ND-2000c spectrophotometer. Fragmented and labeled ss-cDNA was hybridized for 16 h at 45 °C on GeneChip Clariom S Array Rat (Thermo Fisher Scientific, No. 902921). GeneChips were washed and stained in the GeneChip Fluidics Station 450. GeneChips were scanned using the GeneChip Scanner 3000 7G. The gene expression profile of the masseter muscle between the suture and control groups (two samples each) and between the suture and cut groups (two samples each) were evaluated using Transcriptome Analysis Console (TAC) Software (|log FC| > 2 and *p* < 0.05, Thermo Fisher Scientific). The PPI network was visualized using the STRING (Search Tool for the Retrieval of Interacting Genes/Proteins) database (version 11.5 https://string-db.org, accessed on 14 July 2022) [22]. A confidence score of >0.4 was set as the cutoff. Cytoscape software (version 3.9.1, The Cytoscape Consortium, San Diego, CA, USA) was used, and the functional modules of the PPI networks were detected using the Molecular Complex Detection plug-in (MCODE; version 2.0.0, The Cytoscape Consortium) of Cytoscape. In addition, the Cytohubba plug-in (version 0.1, The Cytoscape Consortium) of Cytoscape software identified the hub genes.

## 5. Conclusions

In this study, the muscle fiber composition of the masseter muscle appeared to change from a fast-twitch glycolytic type (type 2B) and the intermediate type (type 2X) to a slow-twitch type (type 1) by changing the innervating nerves from masseteric nerves to hypoglossal nerves. Although limitations such as the small sample size remain, these results might support the validity of the dual innervation technique for the surgical treatment of facial paralysis to maintain resting tonus as well as movement, contributing to more natural facial expression.

## Figures and Tables

**Figure 1 ijms-23-07856-f001:**
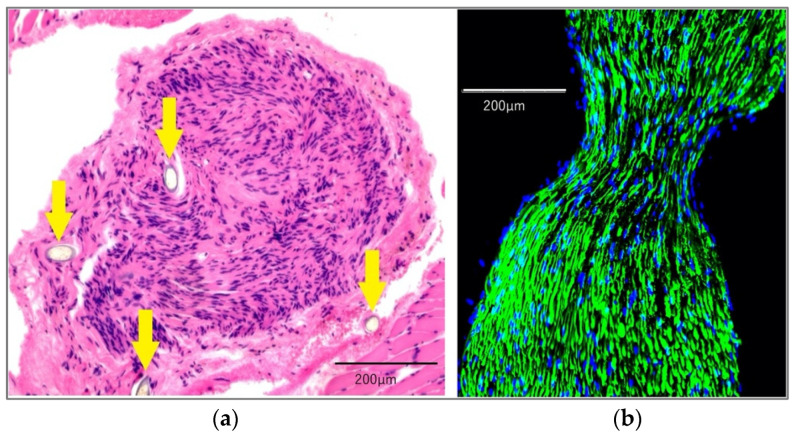
Histological findings of nerves in the suture group. (**a**) The figure shows the distal stump of the masseteric nerve; it was sutured with the hypoglossal nerve. The yellow arrows indicate the nylon sutures used for coaptation. There were no morphologically abnormal findings. These were stained with H&E. (**b**) Immunofluorescence staining of the coaptation area of the hypoglossal nerve and masseteric nerve. All nerves, including the coaptation area, were positive for neurofilament antibodies. Representative immunostaining with neurofilament antibodies (green) was performed using 4′,6-diamidino-2-phenylindole (DAPI; blue).

**Figure 2 ijms-23-07856-f002:**
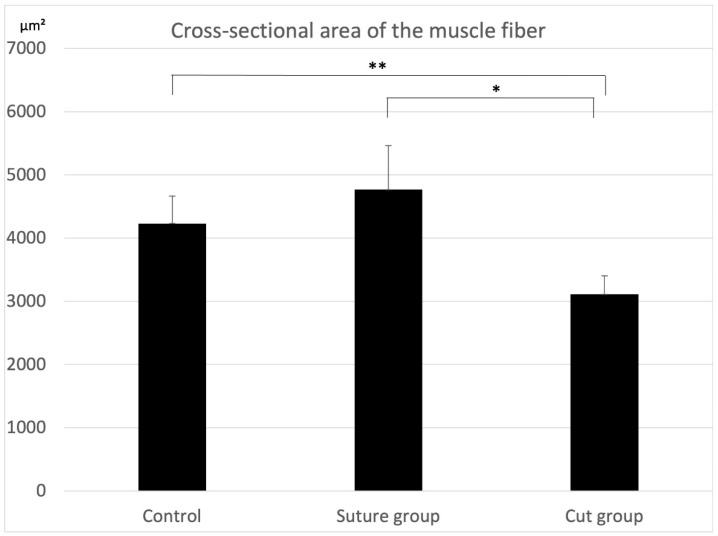
Four 1000 × 1000 µm^2^ HE-stained images of cross-sectional areas of muscle fibers were chosen from each specimen of the control, cut group, and suture groups. The average cross-sectional area of muscle fibers was 4231.20 ± 432.70 µm^2^ (mean ± standard deviation; SD) in the control, 4767.32 ± 695.04 µm^2^ in the suture group and 3112.00 ± 293.49 µm^2^ in the cut group. The area of muscle fibers in the cut group was significantly smaller than that in the other groups (*n* = 4 each, * *p* < 0.01, ** *p* < 0.05; one-way analysis of variance (ANOVA)).

**Figure 3 ijms-23-07856-f003:**
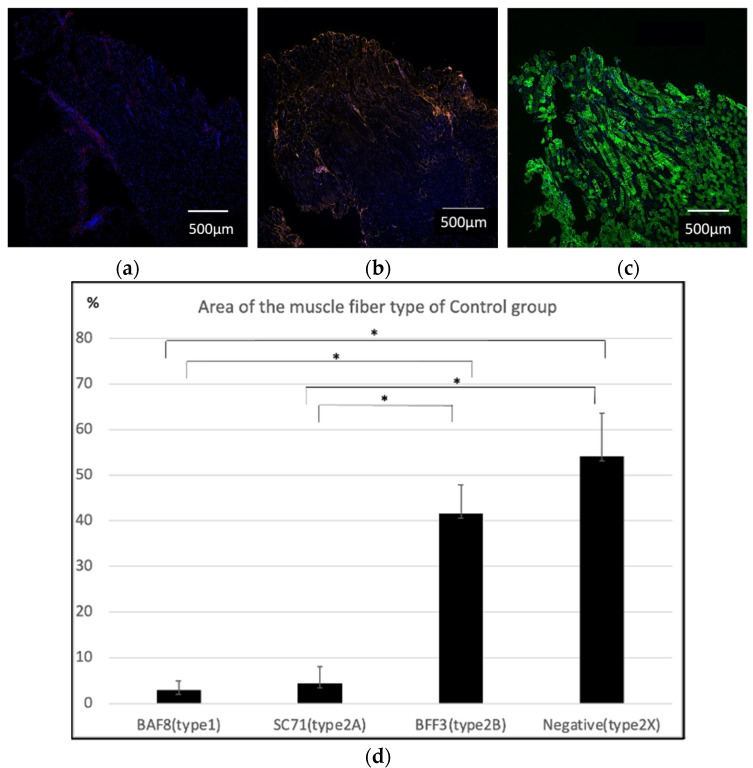
Immunohistochemistry of the masseter muscle in the control group. (**a**) Anti-BAF8 (red) was used for the staining of type 1 muscle fibers, and almost all specimens were negative. (**b**) Anti-SC71 (orange) staining was performed to detect type 2A fibers, and a slightly positive area was observed. (**c**) Anti-BFF3 (green) staining suggested that most of the masseter muscle fibers were composed of type 2B fibers. Nuclear counterstaining was performed using DAPI (blue). (**d**) Anti-BAF8-, anti-AC71-, and anti-BFF3-positive areas were analyzed with four specimens. The average percentage area of type 1, type 2A, type 2B and the estimated type 2X fibers were 2.92 ± 2%, 4.40 ± 3.62%, 41.52 ± 6.30% and 51.16 ± 6.77%, respectively. This indicated that the original masseter muscle was composed mainly of type 2 fibers, especially type 2B fast-twitch glycolytic muscle fibers (*n* = 4 each, * *p* < 0.01, one-way ANOVA).

**Figure 4 ijms-23-07856-f004:**
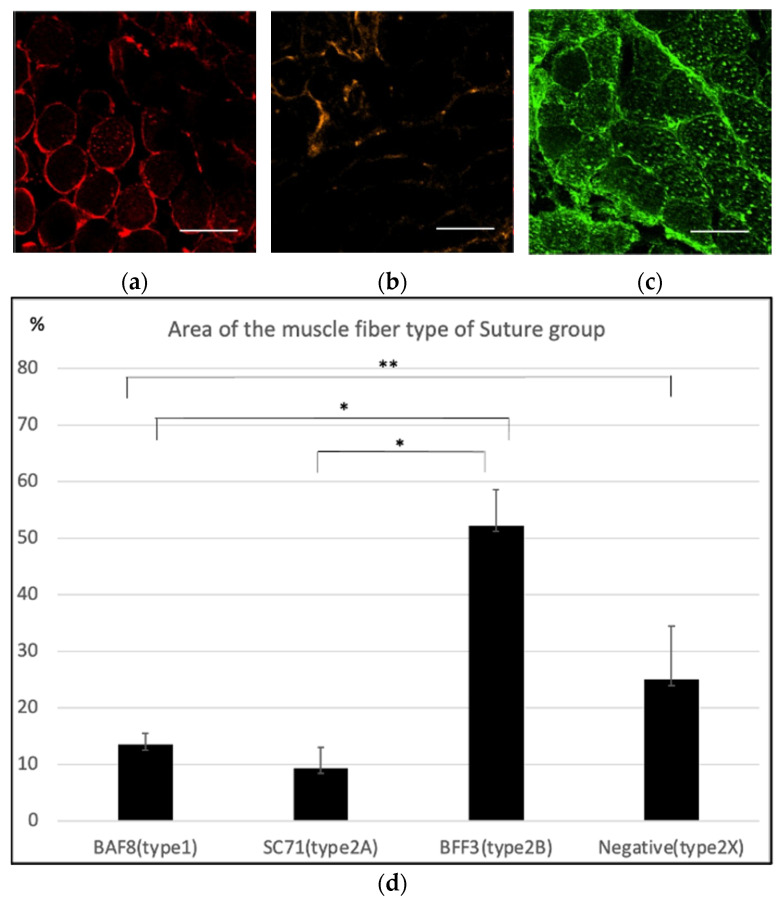
Immunohistochemistry of the masseter muscle in the suture group. (**a**) Anti-BAF8 (red) staining was slightly positive in some areas. (**b**) Anti-SC71 (orange) staining was performed to detect type 2A fibers, and a slightly positive area was observed. (**c**) Anti-BFF3 (green) staining suggested that most of the masseter muscle fibers in the suture group were composed of type 2B fibers. Nuclear counterstaining was performed using DAPI (blue). Scale bar = 100 µm. (**d**) Anti-BAF8-, anti-SC71-, and anti-BFF3-positive areas and the negative area for all antibodies were analyzed with four specimens. The average percentage area of type 1, type 2A, type 2B and the estimated type 2X fibers were 13.50 ± 2.83%, 9.38 ± 2.01%, 52.17 ± 8.30% and 25.00 ± 7.72%, respectively. (*n* = 4 for each group, * *p* < 0.01, ** *p* < 0.05, one-way ANOVA).

**Figure 5 ijms-23-07856-f005:**
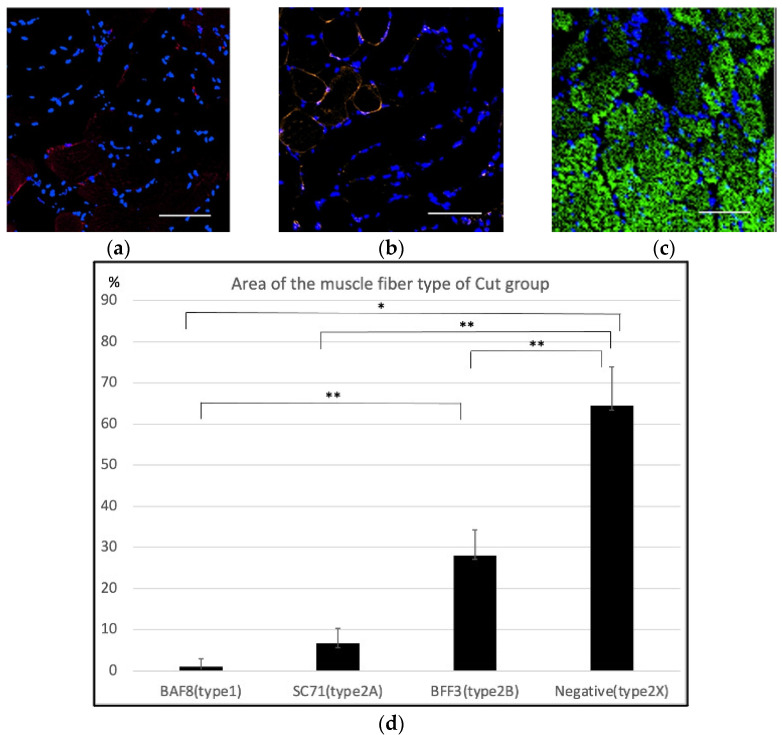
Immunohistochemistry of the masseter muscle in the cut group. (**a**) Anti-BAF8 (red) staining seemed to be almost negative. (**b**) Anti-SC71 (orange) staining showed a slightly positive area. (**c**) Some areas were negative for BAF8, SC71, and BFF3. (**d**) Anti-BAF8-, anti-AC71-, and anti-BFF3-positive areas and the negative area for all antibodies were analyzed with four specimens. The average percentage area of type 1, type 2A, type 2B and the estimated type 2X fibers was 0.92 ± 0.87%, 6.67 ± 6.65%, 27.99 ± 10.80%, and 64.42 ± 7.68%, respectively (*n* = 4 for each group, * *p* < 0.01, ** *p* < 0.05, one-way ANOVA).

**Figure 6 ijms-23-07856-f006:**
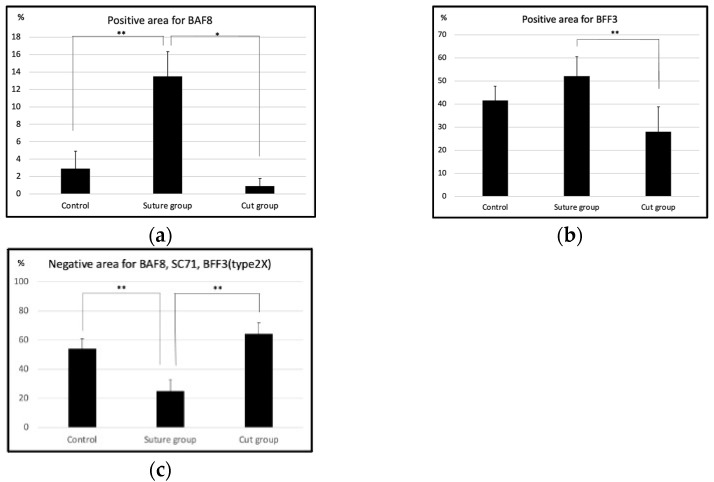
Statistical comparison between the percentage of each muscle fiber type in each group. (**a**) The chart indicates the BAF8-positive areas among the three groups. The suture group showed significantly higher percentages of BAF8 positivity than the other two groups. Its average percentage in the control, suture group, and cut group was 2.92% ± 2.00%, 13.50% ± 2.83%, and 0.92% ± 0.87%, respectively. (**b**) As for BFF3, the cut group revealed significantly lower positivity than the suture group. Its average percentage in the control, suture group, and cut group was 41.52 ± 6.30%, 52.17 ± 8.30%, and 27.99 ± 10.80%, respectively. (**c**) The negative area for BAF8, SC71, and BFF3 was 51.16 ± 6.77% for the control group, 25.00 ± 7.72% for the suture group, and 64.42 ± 7.68% for the cut group. (* *p* < 0.01, ** *p* < 0.05, *n* = 4 for each group, one-way ANOVA).

**Figure 7 ijms-23-07856-f007:**
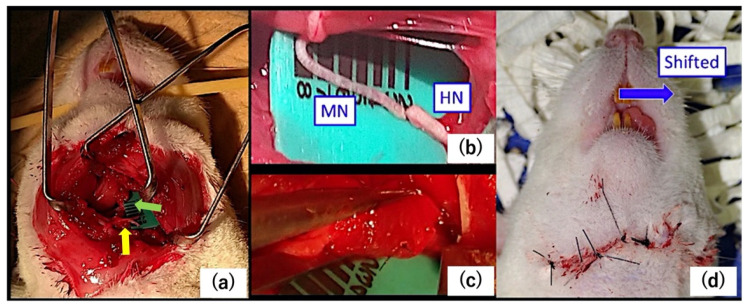
The surgical procedure. (**a**) Weakly magnified surgical photograph. The masseteric nerve (MN: yellow arrow) and the hypoglossal nerve (HN: green arrow) were exposed. (**b**) The MN and the HN were anastomosed. (**c**) The artificial nerve was used as a bridge. (**d**) The tongue was shifted to the direction of the non-operative side (blue arrow) after the surgery.

**Figure 8 ijms-23-07856-f008:**
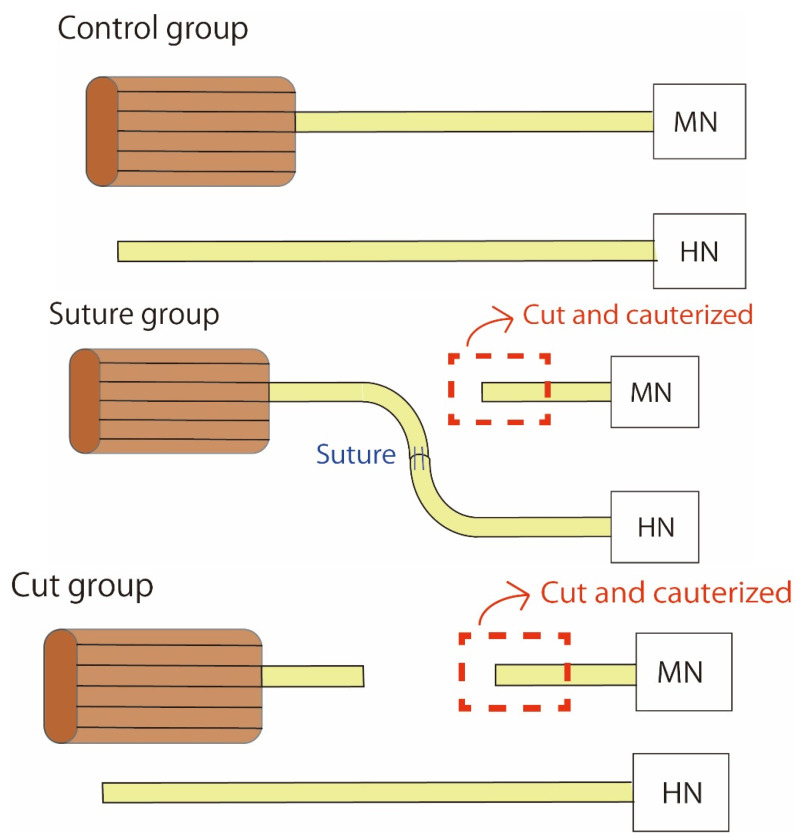
Schema of the experiment. No surgery was performed in the control group. In the suture group, MN was cut, and its distal stump and the proximal stump of HN were sutured. In the cut group, MN was just cut and cauterized. MN: masseteric nerve, HN: hypoglossal nerve.

**Table 1 ijms-23-07856-t001:** Correspondence between the genes identified by Murgia et al. and the genes that were differentially expressed in this study.

Gene	vs. Con	vs. Cut	Gene	vs. Con	vs. Cut	Gene	vs. Con	vs. Cut
Casq2			Asb2			Art3		
Lrrc39			Ppr1r8			Atp1a1		
Lmod2			Ptbp1			Cav1		
Lss			Tfrc			Dysf		
Myom3			Ddah1			Fkbp1a		
Myoz2			Ak1			Gnb1		
Pln			Anxa5			Itgb1		
Tnni1			Epm2a			Jph2		
Sptlc2			Gpd1			Mlec		
Acot9			Gpd2			P4hb		
Ankrd2			Hsp90aa1			Psmd2		
Actn2			Phkb			Rtn4		
Chchd3			Phkg1			Slc37a4		
Ckmt2			Ppkar1a			Smtnl2		
Cpt1b			Psmc4			Tmem43		
Crat			Ryr3					
Decr1			Fndc1					
Ech1			Pgk1					
Hadh			Tnc					
Mrps36			Tpi1					
Mrps7			Agl					
Ndufs7			Ahnak2					

In the gene name tab, genes involved in oxidative muscle are colored blue, and those involved in glycolytic muscle are colored orange. The vs. con column is colored pink if the gene in the suture group is upregulated relative to the control group expression and blue if the gene is downregulated relative to the control group expression. The vs. cut column is colored yellow if the gene in the suture group is highly expressed relative to the control group expression and green if the gene is downregulated relative to the control group expression.

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
