# Peer review of "Muscle Fiber Composition Changes after Selective Nerve Innervation"

_ijms, 2022, doi:10.3390/ijms23147856_

Round 1

Reviewer 1 Report

This study potentially provides important information of an already validated approach to treat facial paralysis. However I have a number of major issues with the presentation and design of the study.

- The results are poorly written. The histological results could be further described and elaborated to clearly articulate what was found. There is not mention of replication and diagrams are poorly labelled and figure legends do not adequately describe what is featured. There are no scale bars and some images are extremely dull, making it difficult to see what is depicted.

- The study is poorly replicated and with a sample size of 2 per condition, how can the authors be sure that their expression results are truly representative? 

- Grammar needs to be corrected throughout. There are many incomplete sentences (ie. Abstract Line 24-25 "Microarray....Perm1".

Author Response

Dear Reviewer 1:

Thank you for your insightful review. We have revised the manuscript based on your comments. Please find below our point-by-point responses to all your comments.

We deeply appreciate your thorough review and look forward to receiving your response regarding our revised manuscript.

---------------------------------------------------------------------------------------------------

Reviewer ’s Comments and Suggestions for Authors

  1. The results are poorly written. The histological results could be further described and elaborated to clearly articulate what was found. There is not mention of replication and diagrams are poorly labelled and figure legends do not adequately describe what is featured. There are no scale bars and some images are extremely dull, making it difficult to see what is depicted.

Thank you for your suggestion. We have replaced the unclear figures with appropriate figures and added scale bars for all figures. We have also provided a detailed explanation of the figures as well as performed quantitative analysis on the data of muscle fiber areas themselves and antibody-positive areas in multiple specimens.

Based on the above revisions, corrections and additions were made in the Results section.

  1. The study is poorly replicated and with a sample size of 2 per condition, how can the authors be sure that their expression results are truly representative?

Thank you very much for this important insight. To ensure the success of this experiment, we repeated the same surgical procedure more than 10 times as a preliminary experiment. This surgical procedure was minimally invasive on the rats, requiring only approximately 1 h of surgery. We also believe that the reproducibility of this surgical procedure is high.

The reproducibility of this immunohistological study was demonstrated on the basis of the results of the analysis of antibody-positive areas in multiple specimens.

Although the number of microarray specimens used in this study was small, as the reviewer pointed out, it served as a preliminary experiment to understand the overall trend. We plan to conduct a similar study with a larger number of specimens in the future.

The Introduction, Results, and Discussion sections were revised to incorporate these points.

  1. Grammar needs to be corrected throughout. There are many incomplete sentences (ie. Abstract Line 24-25 "Microarray....Perm1".

Thank you for pointing this out. We apologize for all errors related to English grammar despite having our manuscript edited by a native English speaker via an English proofreading service before submission. The manuscript was proofread in English again.

Reviewer 2 Report

The authors studied the changes of gene expression and myosin heavy chain expression in masseter muscle of rat after reinnervation. The manuscript is interesting, however, I have a few objections:

1. The authors did not quantitatively assess the proportions of different muscle fibre types in different muscles. This analysis should be added to the results. Moreover, they did not check for the presence of type 2x fibres. Since the transition of muscle fibre type is usually slow and can go through hybrid fibres, the analysis of hibrid fibres is also warranted (DOI: 10.1007/s004240100002).

2. The authors did not perform the morphometric analysis of muscle fibres (diameter, cross-sectional area etc.), which especially in rodents is in correlation with metabolic and functional properties of muscle fibres (DOI: 10.1007/s00418-019-01810-7). Could this analysis be added?

3. The beginning of the result section is not clear, please expand this section.

4. The immunofluorescent figures are not discernible well, can the contrast of color be increased?

5. How was the statistical analysis performed? 

6. Did you find any changes in gene expression that could be linked to changes in capillarization after de and reinervation? Please discuss.

Author Response

Dear Reviewer 2:

Thank you for your insightful review. We have revised the manuscript based on your comments. Please find below our point-by-point responses to all your comments.

We deeply appreciate your thorough review and look forward to receiving your response regarding our revised manuscript.

---------------------------------------------------------------------------------------------------

Reviewer ’s Comments and Suggestions for Authors

  1. The authors did not quantitatively assess the proportions of different muscle fibre types in different muscles. This analysis should be added to the results. Moreover, they did not check for the presence of type 2x fibres. Since the transition of muscle fibre type is usually slow and can go through hybrid fibres, the analysis of hibrid fibres is also warranted (DOI: 10.1007/s004240100002).

Thank you for your suggestion. In this study, we calculated the positive area ratio of each antibody in immunohistologically processed specimens and quantitatively evaluated the ratio of muscle fiber types. As a result, we were able to present data supporting the changes in muscle fiber types in the three groups.

In addition, we read the paper you kindly told us. We totally agree that a discussion on type 2X fibers is important. However, in this study, only the muscle fibers that are not positive for any antibody can be recognized as type 2X muscle fibers. Because of the possibility of the existence of hybrid muscle fibers, this may not provide reliable data. Following your suggestion, a discussion on type 2X fibers was added in the manuscript.

  1. The authors did not perform the morphometric analysis of muscle fibres (diameter, cross-sectional area etc.), which especially in rodents is in correlation with metabolic and functional properties of muscle fibres (DOI: 10.1007/s00418-019-01810-7). Could this analysis be added?

Thank you very much for your comment. We have added the analysis of the cross-sectional area of each muscle fiber. The results of this analysis revealed that there was no significant difference in the area of the muscle fibers between the suture group and control group; however, the average area in the cut group was significantly smaller than that in the other groups due to muscle atrophy. Moreover, we also analyzed the percentage of muscle fiber areas positive for each antibody and found that the muscle fibers in the suture group showed significant positivity for BAF8 antibodies, indicating the transition of the fibers from a fast-twitch to a slow-twitch type.

  1. The beginning of the result section is not clear, please expand this section.

Thank you for the suggestion. We revised the Result section accordingly and moved the first line of the Result section to the Materials and Methods section because its location was inappropriate.

  1. The immunofluorescent figures are not discernible well, can the contrast of color be increased?

We apologize for the inconvenience. We have revised the figures with better resolution.

  1. How was the statistical analysis performed? 

We have revised the Materials and Methods section to explain this point. For gene expression and KEGG pathway analysis, statistical analysis was performed using TAC and STRING applications.

  1. Did you find any changes in gene expression that could be linked to changes in capillarization after de and reinnervation? Please discuss.

Thank you for this suggestion. Interestingly, angiomotin (Amot), which plays an important role in the directional migration of capillaries, was upregulated in both the suture and control groups. However, this change was considered to be due to vascular regeneration after surgery. In other words, during surgery, the masseteric nerve and the hypoglossal nerve were dissected from the surrounding area about 1–1.5-cm long from the transected edge before suturing, which temporarily removed the nutrient capillaries. The subsequent invasion of capillaries from the surrounding area was expected and was thought to be responsible for the elevated Amot.

We truly consider that capillarization is one of the keys to revealing the nature of de- and reinnervation. We would like to reveal the interaction between capillarization and muscle fiber types by developing a device on the operation method in future studies.

Round 2

Reviewer 1 Report

The manuscript is improved from the initial version, however there are still some concerns that I have about the validity of the microarray data. 

Considering the microarray results are taken from just two samples from each group, I would like to see the raw data to determine how close the two replicates (for each sample) lie to each other. If there are huge differences, at least a third replicate is needed to validate these differences.

Author Response

Dear Reviewer 1:

Thank you for your insightful review. We have prepared the necessary data based on your comments. Please find below our response to your comments.

We deeply appreciate your thorough review and look forward to receiving your response regarding our revised manuscript.

---------------------------------------------------------------------------------------------------

Reviewer ’s Comments and Suggestions for Authors

Considering the microarray results are taken from just two samples from each group, I would like to see the raw data to determine how close the two replicates (for each sample) lie to each other. If there are huge differences, at least a third replicate is needed to validate these differences.

Thank you for your suggestion. We attached the raw data of the microarray and would appreciate it if you kindly check it.

Reviewer 2 Report

The authors satisfactorily addressed some of my objections. The following still need to be improved:

1. According to the study DOI: 10.1007/s004240100002, rat masseter muscle is predominantly composed of type 2x fibres. Therefore, analysis of type 2x fibres is mandatory to get reliable results.

2. In the result and discussion section, you write about the percentage of BAF8 positive areas. What did you measure? number of fibres? areas?

3. In figure captions there is written that the paired t test was used for statistical analysis, however, in the methods section you claim you used one-way ANOVA?

4. You report that the average fibre area is 5785.55 ± 344.14 μm2. This seems too large. It would mean that the average fibre has 76 um in diameter, however, usual fibres are around 40 um in diameter. How was the area measured? DOI: 10.1007/s004240100002

Author Response

Dear Reviewer 2:

Thank you for your insightful review. We have revised the manuscript based on your comments. Please find below our point-by-point responses to all your comments.

We deeply appreciate your thorough review and look forward to receiving your response regarding our revised manuscript.

---------------------------------------------------------------------------------------------------

Reviewer ’s Comments and Suggestions for Authors

  1. According to the study DOI: 10.1007/s004240100002, rat masseter muscle is predominantly composed of type 2x fibers. Therefore, analysis of type 2x fibers is mandatory to get reliable results.

Thank you for your suggestion. According to Bottinelli et al. (DOI: 10.1113/jphysiol.1991.sp018617), “monoclonal antibodies cannot detect the co-existence of type 2A with 2X or of type 2X with 2B MHC, as none of the antibodies specifically stains type 2X MHC. Identification of type 2X MHC is obtained by negative staining with the other antibodies used”. As an approximation of type2X expression, the sum of the average negative area percentages for BAF8, SC71, and BFF3 antibodies was calculated in 4 slices. The results were 51.16% ± 6.77 for the control group, 25.00% ± 7.72 for the suture group, and 64.42 ± 7.68 for the cut group and the percentage was significantly lower in the suture group. We estimated that the conversion of type 2X to type 1 had occurred following the cross innervation. We revised the Discussion section and added the article you kindly suggested to us to the Reference. (Line 88-100, 179-180, 204-205, 219-220, 252-265, 366-368)

  1. In the result and discussion section, you write about the percentage of BAF8 positive areas. What did you measure? number of fibres? areas?

Thank you very much for your comment. We calculated the area of the muscle fibers positive for each antibody in each 1000 x 1000µm2 image. We revised the figure captions. (Line 179, 193, 204)

  1. In figure captions there is written that the paired t test was used for statistical analysis, however, in the methods section you claim you used one-way ANOVA?.

Thank you for the suggestion. Since this is an analysis of a continuous variable among three groups, we used one-way ANOVA and performed the Tukey Kramer test for the post hoc test. We revised the captions of figure2, 3d, 4d, 5d and 6.

  1. You report that the average fibre area is 5785.55 ± 344.14 μm2. This seems too large. It would mean that the average fibre has 76 um in diameter, however, usual fibres are around 40 um in diameter. How was the area measured? DOI: 10.1007/s004240100002?

Thank you for the important comment. Considering your suggestion, we re-cut the frozen specimen perpendicular to the muscle fibers and examined it with HE staining. As a result, the average area of muscle fibers was 4231.20 ± 432.70 µm2 (mean ± standard deviation; SD) in the control, 4767.32 ± 695.04 µm2 in the suture group and 3112.00 ± 293.49 µm2 in the cut group. In the study you mentioned, they use female Wister rats, on the other hand, in this study, we used male SD rats. According to the previous studies performed by Massarelli et al. of Charles River Laboratories (www.criver.com) and Hayakawa et al. (DOI: https://doi.org/10.2131/jts.38.855), the average body weight in 18 weeks of the male SD rats was almost 3 times heavier than that of the female Wister rats. Taking it into account, we consider that the fact that the areas of the muscle fibers were greater than those in the reference article does not contradict the facts. I attached the 2 PDF files as references.

We revised line 76-77, 163-165 and 337-338 according to the new results.

Round 3

Reviewer 1 Report

I am satisfied that most of the differences observed in the microarray are replicated between samples.

I would suggest she minor formatting of figures to ensure they are the same size and I would also label the axes of all graphs.

Reviewer 2 Report

The authors satisfactorily addressed most of my comments.

1. In the study you mentioned, they use female Wister rats, on the other hand, in this study, we used male SD rats. According to the previous studies performed by Massarelli et al. of Charles River Laboratories (www.criver.com) and Hayakawa et al. (DOI: https://doi.org/10.2131/jts.38.855), the average body weight in 18 weeks of the male SD rats was almost 3 times heavier than that of the female Wister rats. Taking it into account, we consider that the fact that the areas of the muscle fibers were greater than those in the reference article does not contradict the facts. I attached the 2 PDF files as references.

In my experience the diameter of muscle fibres does not so significantly vary with body weight. For example, even muscle fibre diameters in healthy mice and rats are very similar. Even in humans, the fibres are only slightly larger. Since oblique sectioning could be a problem as you noticed, that is why it is usual that minimal Feret or average Feret diameter is reported and taken as a main measure for muscle fibre size. Minimal Feret diameter and average Feret diameter are least prone to sectioning angle. See: https://doi.org/10.1111/jmi.12985

Please calculated Feret diameters or discuss this limitation in discussion section.